# Enhancing surveillance of sexually transmitted infections in England with gender identity and behavioural data: The GUMCAD STI Surveillance System

Hamish Mohammed[1]*, Stephanie J. Migchelsen[1], Stephen Duffell[1], Ana Karina Harb[1], Tika Ram[1], John Were[1], Sheel Patel[2], Monty Moncrieff[3], James Hardie[2], Maryam Shahmanesh[4], David Phillips[5], Sonali Wayal[4,6], Anthony Nardone[1], Ann Sullivan[2], Claudia Estcourt[7], Jackie A. Cassell[1,8,9], Gwenda Hughes[10], Katy Sinka[1]

1 United Kingdom Health Security Agency, London, United Kingdom, 2 Chelsea and Westminster Hospital NHS Foundation Trust, London, United Kingdom, 3 London Friend, London, United Kingdom, 4 University College London, London, United Kingdom, 5 Croydon University Hospital NHS Trust, London, United Kingdom, 6 Heidelberg Institute of Global Health (HIGH), Faculty of Medicine and University Hospital, Heidelberg University, Heidelberg, Germany, 7 Glasgow Caledonian University, Glasgow, United Kingdom, 8 Brighton and Sussex Medical School, Brighton, United Kingdom, 9 University of Kent, Canterbury, United Kingdom, 10 London School of Hygiene and Tropical Medicine, London, United Kingdom

* hamish.mohammed@ukhsa.gov.uk

**Editor:** Sebastian Suarez Fuller, University of Oxford Nuffield Department of Clinical Medicine: University of Oxford Nuffield Department of Medicine, UNITED KINGDOM OF GREAT BRITAIN AND NORTHERN IRELAND

## Abstract

### Background

There has been an increasing trend in bacterial sexually transmitted infections (STIs) in England since the early 2000s. Since 2008, surveillance of STIs in England has been conducted using the Genitourinary Medicine Clinic Activity Dataset (hereafter referred to as 'GUMCAD'), a depersonalised dataset of all attendances at all publicly-commissioned sexual health services (SHS). The aim of is this article is to describe and evaluate the piloting and rollout of an enhanced specification of GUMCAD at SHS in England.

### Methods

GUMCAD was enhanced in 2019 to allow SHS to report the gender identity (whether cisgender, transgender, gender diverse) of service users, and selected behavioural information collected during routine sexual history-taking such as the number of recent sex partners (last 3 months).

### Results

Feasibility and acceptability of reporting these new data were confirmed in the pilot stage. 2023 was the first year over which most SHS (93%, 224/241) submitted enhanced GUMCAD data. Of all 4,610,410 consultations at SHS in 2023, gender

**Data availability statement:** The data that support the findings of this article have been assessed by the UK Health Security Agency's Office for Data Acquisition and Release as having sensitive personal information and are therefore not publicly available to protect participant privacy. However, some aggregate data may be available upon reasonable request from the UKHSA. Requests can be directed to DataAccess@ukhsa.gov.uk.

**Funding:** Public Health England, the predecessor to UKHSA, funded the pilots, and the salaries of all coauthors employed by PHE at that time. EpiConcept and Parexel International provided support in the form of salaries for authors AN and JW respectively at the time of submission in May 2025, but did not have any additional role in the study design, data collection and analysis, decision to publish, or preparation of the manuscript.

**Competing interests:** AN was a member of the Steering Group for the pilots of the enhanced GUMCAD specification while he was an employee of Public Health England (PHE; the predecessor to the UK Health Security Agency [UKHSA]); he is now an employee of EpiConcept, a privately funded company. JW conducted analyses of the data collected from the pilots (the results of which are presented in the Appendix) while he was an employee of PHE; he is now an employee of Parexel International Limited, a privately funded clinical research organisation. None of the other coauthors reported any conflicts of interest. Both AN and JW were employed by PHE at the time of the pilot. The specific roles of these authors are articulated in the 'author contributions' section. This does not alter our adherence to PLOS ONE policies on sharing data and materials.

identity (96% of consultations) and data on whether this varied from the sex registered at birth (88% of consultations) were well reported. Transgender women, transgender men, and gender-diverse individuals (identifying as non-binary or in any other way) respectively comprised 0.4%, 0.4%, and 0.5% of all consultations in 2023. There was less complete reporting of recent sex partners but, where reported, gay, bisexual and other men who have sex with men were more likely to report multiple recent sex partners.

## Conclusions

This enhancement provides novel insights into sexual health need relevant to targeting existing and novel preventative interventions for STIs such as 4CMenB vaccination for gonorrhoea and doxycycline post-exposure prophylaxis (doxyPEP) for syphilis in England. The reporting of these new STI surveillance data also raise new complexities in interpretation, and behavioural data completeness will require further support and development.

## Introduction

Sexually transmitted infections (STIs) include a range of bacterial, viral and protozoal infections which are primarily transmitted through sexual contact. STIs can result in acute symptoms such as genital ulcers or discharge, dysuria and testicular or pelvic pain; while many people may not experience any symptoms, if untreated, STIs may result in further onward transmission and adverse sequelae such as ectopic pregnancy, infertility, and neonatal mortality.

In England, there has been an increasing trend in diagnoses of bacterial STIs such as gonorrhoea and syphilis since the early 2000s [1,2]. When compared to heterosexual people, gay, bisexual and other men who have sex with men (GBMSM) have disproportionately higher diagnosis rates of gonorrhoea and syphilis [3]; this may be a result of factors including higher rates of STI testing as well as behaviours such as multiple condomless sex partners [4,5]. Since the cessation of the last national lockdown to control COVID-19 in July 2021, there was a rapid resurgence in gonorrhoea [6] particularly among heterosexuals [7].

Genitourinary medicine, also known as Sexual Health and HIV Medicine, is a medical speciality in the UK which includes the prevention and management of STIs and HIV, care of people living with HIV, and promotion of overall sexual health and wellbeing [8]. England has a national network of open access sexual health services (SHS) with staff trained in genitourinary medicine who perform STI and HIV testing and management, partner notification (PN), and offer preventative interventions such as HIV pre-exposure prophylaxis (PrEP) [9]. Sexual health services are free at the point of delivery and, relative to other primary care settings, diagnose the majority of STIs [10,11].

In 2008, the Health Protection Agency (now the UK Health Security Agency [UKHSA]) worked with sexual healthcare practitioners to develop and introduce a

new mandatory national STI surveillance system which collected, for the first time, disaggregated data on attendances, diagnoses made and services provided at all SHS in England using the Genitourinary Medicine Clinic Activity Dataset (now referred to as 'GUMCAD') [12]. While GUMCAD has been established as a comprehensive, high-quality STI surveillance system and has been extensively used to inform sexual health policy decisions and evaluate interventions, it was limited by a lack of key information on sexual behaviour and the outcomes of PN. Given the critical role of sexual behaviour and the effectiveness of PN in reducing STI transmission, we introduced an enhancement to the GUMCAD specification including behavioural and PN variables. Additionally, given evidence of higher HIV prevalence among transgender men and particularly transgender women, and high prevalence of other STIs in transgender people (compared to cisgender people) [13,14], UKHSA also added a variable to GUMCAD to assess gender identity (whether cisgender, transgender or gender diverse); this was with the aim of obtaining better data on the epidemiology of STIs in transgender people to support the provision of sexual health services for this population.

In this article, we describe and evaluate the piloting and implementation of this enhancement to the GUMCAD STI surveillance system.

## Methods

UKHSA performs surveillance of STIs, including the piloting described in this article, for health protection purposes under permissions granted to UKHSA to collect and process pseudonymised GUMCAD patient data under Regulation 3 of The Health Service (Control of Patient Information) Regulations 2020, and Section 251 of the National Health Service Act 2006. The analyses presented in this article have been subject to an internal review by UKHSA's Research Support and Governance Office which considered design, content, and feasibility. The review also covered all legal, financial, regulatory, and ethical considerations. As a result of this review, this analysis was categorised as public health surveillance and, as no ethical issues were identified, it was decided that review by an ethics committee would not be necessary. As approved by UKHSA's Caldicott Advisory Panel, a group of information governance experts whose remit includes assessing whether data collections constitute public health surveillance activities, informed consent was not required for the collection of the GUMCAD data used in this analysis.

We conducted a mixed methods iterative pilot of an enhanced GUMCAD surveillance dataset. The legacy GUMCAD specification and its development are described elsewhere [12]. Briefly, GUMCAD is the mandatory surveillance system for STIs in England and is an electronic dataset submitted quarterly to UKHSA by all publicly commissioned SHS since 2008; there were 241 SHS in England registered to report GUMCAD in 2023. SHS, including online services, are commissioned by the 153 upper-tier local authorities (administrative local government bodies) in England. Reporting GUMCAD data to UKHSA for public health surveillance is a mandatory requirement of the Department of Health and Social Care [15]. GUMCAD data are pseudonymised and depersonalised (reported without direct identifiers such as name, date of birth or full address) [16], but have a clinic-specific alphanumeric code (patient ID) that allows follow-up of individuals attending the same service over time. GUMCAD includes demographic information aligned with national data standards [17,18], as follows: gender, sexual orientation, age, ethnicity, country of birth, and lower-layer super output area of residence [small geographical areas with resident populations between 1,000–3,000 people].

To avoid double counting when reporting the numbers of STI tests or diagnoses, episodes of care are created during data cleaning to ensure, for example, only one diagnosis of chlamydia is retained within a 42-day period. A sensitivity analysis comparing 42-days to shorter episode lengths confirmed that reducing this to 14- or 28-days had a negligible impact on the number of episodes [19].

### Development of the enhanced GUMCAD specification

The enhanced GUMCAD specification included behavioural data recommended by the British Association for Sexual Health and HIV (BASHH) for sexual history taking [20] and audits of PN at SHS [21] (Appendix Table 1 in S1 Appendix).

The development of the pilot specification was undertaken by a multi-disciplinary steering group of key stakeholders representing sexual health clinicians and health advisers, academia, LGBTQ+ charities, substance use specialists, software suppliers of electronic patient record [EPR] management systems, and UKHSA.

## Piloting

We performed two pilots, with the findings from the first pilot used to inform refinements to the dataset specification for the second. For the first pilot, a geographically dispersed convenience sample of SHS serving diverse communities including GBMSM, young adults, and minority ethnic populations was recruited. When selecting SHS for the pilot, we prioritised those using the most commonly used EPR software at SHS in England (at least one SHS in each of the 9 public health regions of England), and contacted them directly to seek their participation. Recruitment of SHS to the first pilot commenced in early 2013, with staggered start dates. The pilot was run for 8-weeks at each SHS and included all service users. Technical guidance on recording each new variable was provided to all pilot SHS and their software providers to standardise data collection. Prior to data collection, UKSHA staff held initiation meetings with participating services to explain the specification and pilot methodology. Each of the pilot SHS' EPR software suppliers was required to make changes to their service's EPR system to facilitate the recording, extraction and uploading of data (as a single comma-separated value file) using the updated GUMCAD specification to the established secure online facility for GUMCAD surveillance reporters to submit data to UKHSA [22]. With the endorsement of the steering group, a second pilot using a revised dataset specification was conducted in 2015/16 in 5 of the 8 SHS which participated in the first pilot.

## Assessing the feasibility and acceptability of the enhanced GUMCAD specification

Firstly, we assessed the feasibility and acceptability of the pilot enhancement to service providers (doctors, nurses and health advisers [healthcare practitioners with expertise in PN [23]) at pilot SHS using self-administered, anonymous online surveys conducted after each of the 2 pilots (Appendix Figure 1 in S1 Appendix), and confidential key informant interviews using a topic guide after the first pilot (Appendix Document 1 in S1 Appendix).

The survey included close-ended questions on the acceptability, ease and potential benefit of collecting additional behavioural data and how easily this could be integrated into routine clinical practice. After the first pilot, respondents indicated their willingness to participate in confidential key informant interviews. The aim was to interview at least one member of staff at each of the SHS in the first pilot or until thematic saturation was achieved. We transcribed the interviews into anonymised scripts, then performed thematic analysis of coded interview data on the feasibility and acceptability of collecting the enhanced data after the first pilot.

Secondly, we performed a descriptive analysis of the data on service users reported by pilot SHS to assess the representativeness of service users with enhanced GUMCAD data submitted (versus those without), and item non-response (all variables reported with a 'not known' or 'not stated' response) using proportions; these analyses were conducted on pseudonymised GUMCAD data extracted on 07/11/2016. As the anonymous survey data from service providers and data quality of the data on service users from the second pilot were used to inform the final version of the enhanced GUMCAD specification, we present the results of descriptive analyses of survey responses and data quality indicators (frequencies and proportions) from the second pilot. The authors did not have access to information that could identify individual survey participants or service users in GUMCAD during or after data collection.

The steering group endorsed the final version of the enhanced GUMCAD specification, after which UKHSA sought approval for this specification from the national Data Coordination Board (DCB) as the new mandatory STI surveillance dataset in England. The DCB assessed the burden and feasibility of data collection at SHS, as they do for other publicly provided healthcare settings.

## Results

### Feedback from the first pilot

Eight SHS participated in the first pilot in 2013/14. These 8 services were dispersed geographically throughout England (the East of England, London, Midlands, Southeast and Southwest; Appendix Table 2 in S1 Appendix). All participating SHS successfully submitted their piloted GUMCAD enhanced data through the UKHSA's secure online facility.

In the first pilot, a total of 4 members of staff from 3 of the participating SHS was interviewed. They all agreed that the proposed questions in the enhanced GUMCAD specification were asked routinely during sexual history taking and would be beneficial to the management of service users and to public health. However, the level of detail in the initially proposed specification in 2013 was considered overly burdensome, beyond the scope of routine practice, and disproportionate relative to the prevalence of certain behaviours, such as chemsex, among all SHS attendees. Despite this, one SHS continued to use that specification after the pilot while others used specific sections such as recreational drug use behaviours after the 8-week pilot period for internal use. The feedback from the first pilot was used to refine the enhanced GUMCAD specification to remove detailed questions about sexualised drug use and some GBMSM-specific sexual behaviours, then a revised specification was used in the second pilot. Revisions to the specification were made by a consensus among members of the steering group after the review of the qualitative (key informant interviews) data summarised above, and quantitative data on item non-response for the enhanced GUMCAD data from service users.

### Feedback and data quality from the second pilot

Five of the 8 SHS which participated in the first pilot also participated in the second in 2015/16, all of which submitted these data through the UKHSA's secure online facility. Among the 23,107 records submitted by these 5 SHS over the 8-week pilot period (Appendix Table 2 in S1 Appendix), 94.3% (21,789) matched a record in their routine GUMCAD extract; this ranged from 78.8% to 99.9% among the pilot SHS. The distribution of gender, age-group and ethnicity was similar when comparing records for which enhanced data were submitted to other records (Appendix Figures 2-4 in S1 Appendix).

A total of 21 clinic staff from 4 pilot SHS completed the online survey. Overall, the proposed enhancement to GUMCAD was viewed positively, with 43% of respondents agreeing that the piloted data items were beneficial or consistent with clinical practice for sexual history taking, while the majority (57–71%) agreed that the technical guidance document was clear, and that these data items were easy to collect and acceptable to patients (Appendix Tables 3-4 in S1 Appendix).

### Finalising the specification for approval

While the second pilot demonstrated that the proposed enhancement to GUMCAD was feasible to collect and acceptable within routine clinical practice from the perspective of service providers, further refinements were made to the final specification for simplicity and to reduce the burden of data collection before applying for DCB approval. For example, collecting data on whether recent sexual intercourse took place outside the UK was removed due to the difficulties assessing this reliably, while data on past STI diagnoses or HIV testing were removed due to high item non-response, but also because this information would be available in GUMCAD for people reattending the same SHS.

To enable the collection of data on gender identity and after input from the steering group, a question to collect data on sex registered at birth was also added to the GUMCAD specification. Gender identity using GUMCAD data is defined using 2 questions: (i) "what is the patient's gender identity?" (female [including transgender women], male [including transgender men], non-binary, other); and (ii) "is the patient's gender identity the same as their sex registered at birth?" (yes, no, not stated). Some additional refinements were made to the GUMCAD specification to account for changes in the delivery of sexual health services taking place at the time – for instance, a new category to monitor testing for STIs or HIV using an online postal self-sampling kit [24] was added to the variable capturing the modality of accessing an SHS (which

had previously only distinguished between face-to-face attendances and telephone consultations). As was the case after the first pilot, the final specification was created by consensus agreement from the steering group.

The DCB granted approval in 2018 [25] and UKHSA commenced rollout in 2019. Following a pause in 2020 during the COVID-19 pandemic, rollout resumed in September 2021. 2023 was the first calendar year over which more than 90% of SHS reported enhanced GUMCAD data to UKHSA.

### Findings of relevance to improving care

As of December 2023, all 241 SHS in England submitted GUMCAD data to UKHSA, 93% of which submitted data aligned to the enhanced specification of GUMCAD. There was good reporting of gender identity questions, with poorer reporting of data on the number of recent sex partners; these are described below.

Given the gaps in reporting (missing data on sexual behaviour from some SHS in London, as well as online SHS), the first tranche of enhanced GUMCAD data on gender identity and recent sex partners were published as provisional data in July 2024 [26].

Of all the 4,610,410 attendances and consultations at SHS in England in 2023, gender identity was reported for 96% of records, and data on whether this varied from the sex registered at birth was reported for 88% of these records. That year, transgender women (0.4%), transgender men (0.4%) and gender-diverse individuals (0.5%; people identifying as non-binary or in any other way) accessed SHS in England (Fig 1A), with similar proportions among those diagnosed with gonorrhoea (Fig 1B), one of the most commonly reported STIs in England.

In 2023, data on the number of recent (prior 3 months) sex partners was reported for 28% (632,857/2,263,515) of face-to-face attendances in 2023, but these data were not reported for people being tested for STIs using online postal self-sampling kits. In absolute numbers, heterosexual or bisexual women accounted for far more screens and attendances than heterosexual men. Where data on recent partner numbers were available, these women and men typically reported only one sexual partner in the past three months. In contrast, among GBMSM — who more frequently reported multiple partners — most screens and attendances involved men with 2–4 or 5+ recent partners (Fig 2A). The numbers of gonorrhoea and infectious syphilis diagnoses were also highest amongst most gender and sexual orientation groups reporting 1 recent sex partner; however, gonorrhoea and infectious syphilis were most frequently diagnosed in GBMSM reporting two or more recent sex partners (Fig 2B), consistent with the overall profile of clinic attendees in this group.

## Discussion

The pilots of sexual behaviour and gender identity data collection confirmed the feasibility and acceptability of these items, and enabled us to identify the minimum enhanced data that can be collected through routine STI surveillance. A high proportion of SHS reported the enhanced GUMCAD dataset, with better reporting for gender identity than recent sex partners. The provision of open access SHS through a publicly-funded healthcare system using a standardised set of clinical codes to report electronically has facilitated comprehensive reporting of STI surveillance in England, and further development is underway to improve the reporting of gender identity and sexual behaviour data.

UKHSA worked closely with key sexual health stakeholders to develop the enhanced GUMCAD specification which is now the dataset used for mandatory STI surveillance. This, in turn, facilitates the collection of more robust data to assess the uptake and inform the targeting of key interventions such as mpox vaccination among eligible individuals including gender minorities, as well as behavioural data to assess the use of condoms and the frequency of sexualised drug use or multiple partnerships among people attending SHS.

The experience of iteratively piloting the enhanced specification has emphasised the need to balance the granularity of behavioural data with the feasibility and acceptability of collecting and reporting them. A key lever for this enhancement of STI surveillance was the prioritisation of data on sexual behaviours that are routinely assessed during a sexual history at an SHS for inclusion in GUMCAD. Some variables such as self-reported past STI diagnoses or HIV testing were very

 

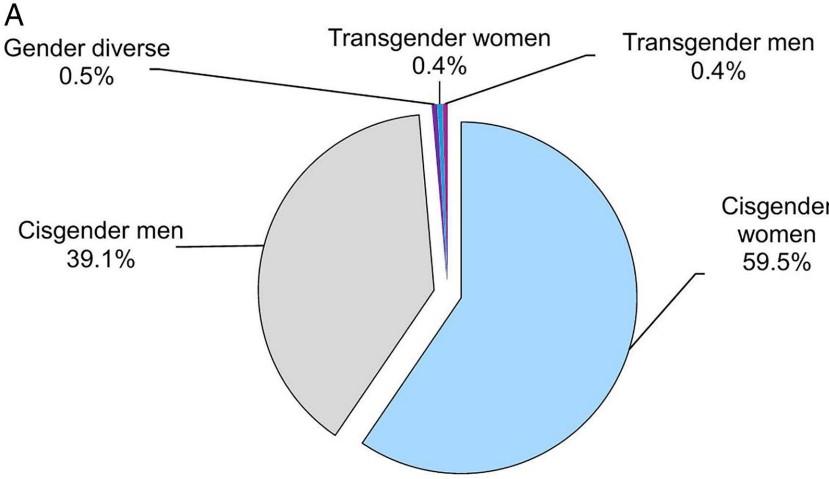

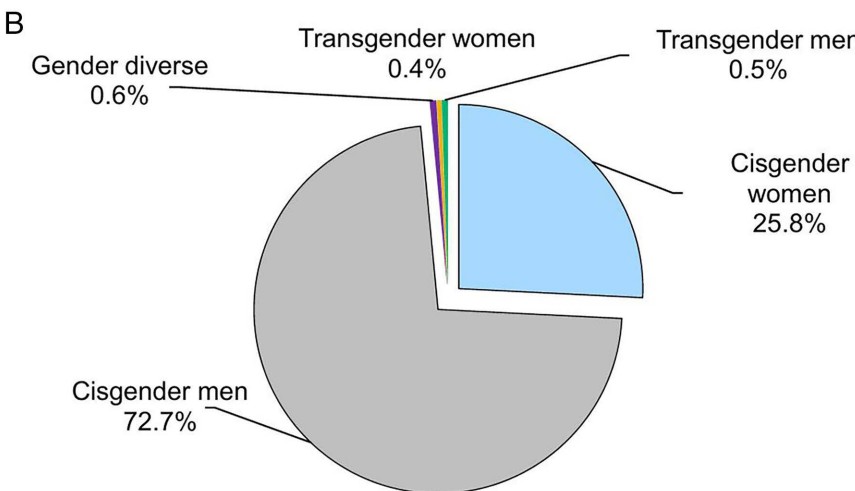

**Fig 1. Number of attendances and consultations (A) and gonorrhoea diagnoses (B) at sexual health services by gender identity, England, 2023.** A. Attendances and consultations at sexual health services by gender identity. B. Gonorrhoea diagnoses at sexual health services by gender identity. Data source: Routine sexual health services' submissions to the GUMCAD Sexually Transmitted Infection Surveillance System. Note: Gender identity was reported by 91% (219/241) sexual health services in England in 2023, representing 88% of the overall number of attendances or consultations at sexual health services in 2023. Cisgender people are those for whom their gender identity is the same as the sex registered at birth. Gender diverse is defined as identifying as non-binary or in any other way.

poorly completed in the pilot and have been excluded from GUMCAD. However, to complement data from GUMCAD, UKHSA regularly runs community surveys to assess SHS access and need and, while these are based on convenience samples, they include data on people who do and those who do not access SHS [27]. UKHSA's collaborators also undertake high quality surveys such as Natsal to collect data from a nationally representative sample of the population [28], and the Gay Men's Sexual Health Survey, a biobehavioural survey of a convenience sample of GBMSM in large English cities [29]. During the pilots, multiple SHS using different EPR software systems were included to ensure that the enhanced specification of GUMCAD was software agnostic, and to facilitate rollout once the specification was approved.

England, in common with other high-income countries, has detected an increase in bacterial STI diagnoses since the end of lockdowns to control COVID-19 in 2021 [30]. England's high STI diagnosis rates compared

A

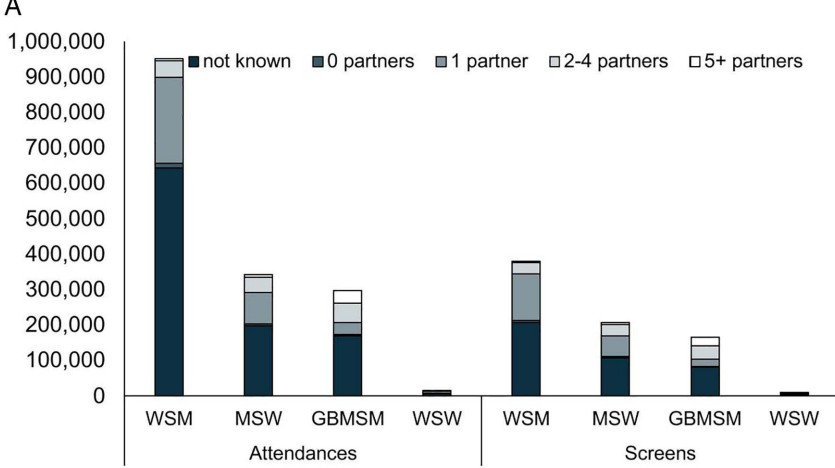

B

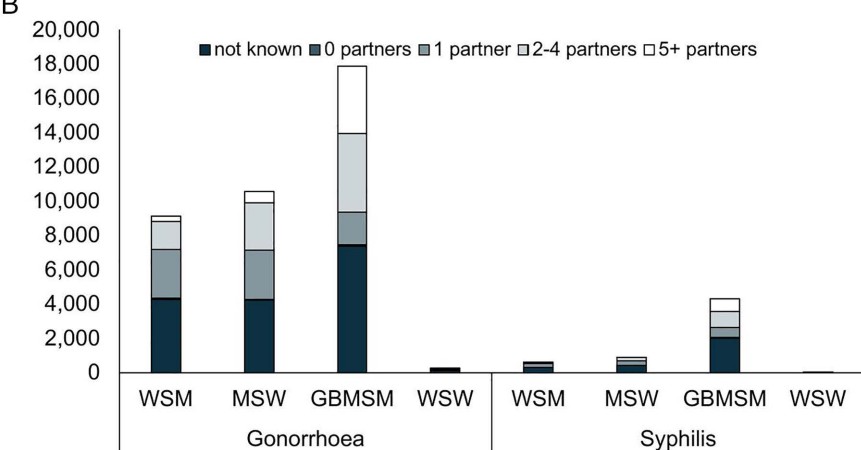

**Fig 2. Number of attendances and sexual health screens (A), and gonorrhoea and syphilis diagnoses (B) at sexual health services by the numbers of recent sex partners (last 3 months), by gender identity and sexual orientation group, England, 2023.** A. Attendances and screens at sexual health services by the numbers of recent sex partners. B. Gonorrhoea diagnoses at sexual health services by the numbers of recent sex partners. Different scales are used on the y-axes of each of the graphs above. Data source: Routine sexual health services' submissions to the GUMCAD Sexually Transmitted Infection Surveillance System. WSM – women who have sex with men; MSW – men who have sex with women (exclusively); GBMSM – gay, bisexual and other men who have sex with men; WSW – women who have sex with women (exclusively). In this figure, 'women' refers to both cisgender and transgender women, and 'men' refers to both cisgender and transgender men. Note: The number of recent sex partners was reported by 88% (206/235) of sexual health services (SHS) in England in 2023. Therefore, the graphs above only include data from the equivalent 88% of SHS – so the numbers of consultations, sexual health screens (tests for chlamydia, gonorrhoea, syphilis or HIV), gonorrhoea diagnosis and infectious syphilis diagnoses are lower than the overall total.

to other countries [31] is in part an artefact of the comprehensive STI surveillance atop a national system for offering high standard and expert-led sexual healthcare. Insights from this enhancement of GUMCAD to contextualise the occurrence of these infections will potentially be of wider benefit to other countries seeing similar patterns in their STI epidemiology. While the vast majority of SHS reported the enhanced specification of GUMCAD in 2023, there were gaps in reporting. Ongoing data quality improvement is underway through the use of a data quality scorecard for SHS, annual data quality meetings with regional UKHSA staff supporting SHS, and bespoke online and in-person seminars to present the enhanced GUMCAD specification and to offer coding advice [32].

Despite the reporting gaps, the distribution of recent sex partners among people attending SHS by gender and sexual orientation reflected that of a large population-based survey, such that GBMSM were more likely to report multiple partnerships than heterosexual men, heterosexual women, or lesbians [4,33]. The relatively low proportion of transgender and gender-diverse people attending SHS may be a reflection of the underlying distribution in the population [34] and the fact that they may access SHS less frequently than cisgender GBMSM (who are recommended to test for STIs quarterly from three anatomical sites if they are at risk of STIs [35]), as well as under-reporting of gender identity data in GUMCAD. The ability to report data on gender identity and whether this varies from the sex registered at birth though GUMCAD presents an opportunity to capture data on whether SHS users are cisgender, transgender or gender diverse then, with more complete reporting of these data, to use these data to more robustly assess inequalities in sexual health.

STI surveillance data, by their very nature, are highly sensitive so UKHSA has published supporting privacy information to explain why STI surveillance data are collected and the confidentiality controls in place to protect privacy [16]. Sexual health services in England operate with unique privacy safeguards and are covered by anonymous access provision in law [36]; in keeping with this, SHS report surveillance data pseudonymously such that individual service users cannot be identified from the information provided. STIs are not legally notifiable in England because the network of open access SHS conducts PN for STIs, and all SHS report depersonalised GUMCAD surveillance data [37,38].

These new enhanced data in GUMCAD are not without limitations, because they only represent people who access a SHS and because this reporting only commenced in most SHS in 2023, so it will take time to build up a time-series of data to contextualise STI trends. Gender identity data were better reported than recent sex partners, possibly because the former may be more routinely collected upon registration of service users; however, further work is needed to minimise item non-response for the enhanced data. We acknowledge the limitation of aggregating cisgender and transgender women in some presentations of GUMCAD data as the route of STI transmission cannot be inferred if transgender women with a penis are classified as WSW. Also, SHS are facing increased demands on the finite appointment time and testing resources, with evidence of unmet need for face-to-face appointments reported by 12% of GBMSM participants of a 2024 community survey [39]. It is therefore important for UKHSA to continue to support SHS to report GUMCAD by making it as easy as possible and making full use of the data to support understanding of STIs and their prevention.

With this enhancement to GUMCAD, UKHSA is now better equipped to understand the factors associated with STI trends, and the gaps in service provision which contribute to poorer control of STIs. Additionally, the ability to assess gender identity in GUMCAD facilitates more robust interpretation of STI surveillance data and a better understanding of the epidemiology of STIs among people who are transgender or gender diverse. However, it does raise emerging complexities of interpretation which will require ongoing reflection and development. The enhanced GUMCAD specification also provides greater scope to use STI surveillance data to inform and evaluate public health interventions such as HIV-PrEP, or to plan for novel ones such as doxycycline post-exposure prophylaxis (doxyPEP) for syphilis prevention and 4CMenB vaccination for gonorrhoea. With triangulation against community survey and other data, there are more complete data to understand the sexual health and wellbeing of the population and to inform the commissioning of SHS and wider sexual health policy to reduce the incidence of STIs and STI-related harm.

## Supporting information

**S1 Appendix. Containing Appendix Table 1–4, Appendix Figure 1–4 and Appendix Document 1.**
(DOCX)

**S1 File. Human Subjects Research Checklist.**
(DOCX)

## Acknowledgments

The authors dedicate this article to the memory of Mr. David Stuart (Chemsex and HIV advocate at 56 Dean St, London – and former member of the Steering Group for the enhanced GUMCAD pilot) and Dr. Helen Wheeler (Clinical Lead and Sexual Health Consultant, and the lead for the enhanced GUMCAD pilot in Bristol).

The authors acknowledge Dr. Alan McOwan (56 Dean St, London); Mrs. Jacinta Ryan (Barnet, London); Dr. Gillian Dean (Claude Nicol Centre, Brighton); Dr. Ian Fairley and Ms. Corinna Dass (YorSexualHealth, York); and Mrs. Tracey Garratt (iCaSH Bedfordshire) for participating in the pilot, and the staff of all pilot services for their contributions. The authors acknowledge Ms. Martina Furegato (a former PHE employee who is now at Oracle Life Sciences) and Dr. Sarika Desai (a former PHE employee who is now at the World Health Organization) for their contributions to the pilot. The authors also thank Ms. Mandy Yung (UKHSA) for her support applying for approval for the enhanced GUMCAD specification.

## Author contributions

**Conceptualization:** Anthony Nardone, Gwenda Hughes.

**Data curation:** Hamish Mohammed, Stephen Duffell, John Were.

**Formal analysis:** Hamish Mohammed, Stephanie J. Migchelsen, John Were.

**Funding acquisition:** Anthony Nardone, Gwenda Hughes.

**Investigation:** Hamish Mohammed, Sheel Patel, Anthony Nardone, Gwenda Hughes.

**Methodology:** Hamish Mohammed, Stephen Duffell, Monty Moncrieff, James Hardie, Maryam Shahmanesh, David Phillips, Sonali Wayal, Anthony Nardone, Ann Sullivan, Claudia Estcourt, Jackie A. Cassell, Gwenda Hughes, Katy Sinka.

**Project administration:** Hamish Mohammed.

**Supervision:** Katy Sinka.

**Writing – original draft:** Hamish Mohammed.

**Writing – review & editing:** Stephanie J. Migchelsen, Stephen Duffell, Ana Karina Harb, Tika Ram, John Were, Sheel Patel, Monty Moncrieff, James Hardie, Maryam Shahmanesh, David Phillips, Sonali Wayal, Anthony Nardone, Ann Sullivan, Claudia Estcourt, Jackie A. Cassell, Gwenda Hughes, Katy Sinka.

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
