## [Decision Letter · Decision Letter 0]

3 Oct 2025

Dear Dr. Mohammed,

Thank you for submitting your manuscript to PLOS ONE. After careful consideration, we feel that it has merit but does not fully meet PLOS ONE’s publication criteria as it currently stands. Therefore, we invite you to submit a revised version of the manuscript that addresses the points raised during the review process.

Please respond fully to each of the reviewers comments in your rebuttal, including the detailed comments of reviewer 1 and the more general comments of reviewer 2, including clarifying the aim of the manuscript and issues around missing data.

We look forward to receiving your revised manuscript.

Kind regards,

Sebastian Suarez Fuller, PhD

Academic Editor

PLOS ONE

Journal Requirements:

The first pilot was funded internally by Public Health England’s Innovation Fund. Public Health England is the predecessor of the UK Health Security (UKHSA). The first author and many coauthors are employees of UKHSA, the national health protection agency for England.

3. Thank you for stating the following in the Competing Interests/Financial Disclosure section:

AN was a member of the Steering Group for the pilots of the enhanced GUMCAD specification while he was an employee of Public Health England (PHE; the predecessor to the UK Health Security Agency [UKHSA]); he is now an employee of EpiConcept, a privately funded company. JW conducted analyses of the data collected from the pilots (the results of which are presented in the Appendix) while he was an employee of PHE; he is now an employee of Parexel International Limited, a privately funded clinical research organisation. None of the other coauthors reported any conflicts of interest.

We note that one or more of the authors are employed by a commercial company: EpiConcept and Parexel International Limited

4. We noted in your submission details that a portion of your manuscript may have been presented or published elsewhere. Please clarify whether this conference proceeding was peer-reviewed and formally published. If this work was previously peer-reviewed and published, in the cover letter please provide the reason that this work does not constitute dual publication and should be included in the current manuscript.

6. Please remove all personal information, ensure that the data shared are in accordance with participant consent, and re-upload a fully anonymized data set.

Additional guidance on preparing raw data for publication can be found in our Data Policy (https://journals.plos.org/plosone/s/data-availability#loc-human-research-participant-data-and-other-sensitive-data) and in the following article: http://www.bmj.com/content/340/bmj.c181.long .

Additional Editor Comments:

Please respond to each of the reviewers comments in your rebuttal.

Reviewers' comments:

Reviewer's Responses to Questions

**Comments to the Author**

1. Is the manuscript technically sound, and do the data support the conclusions?

Reviewer #1: Yes

Reviewer #2: No

2. Has the statistical analysis been performed appropriately and rigorously?

Reviewer #1: N/A

Reviewer #2: N/A

3. Have the authors made all data underlying the findings in their manuscript fully available?

Reviewer #1: No

Reviewer #2: No

4. Is the manuscript presented in an intelligible fashion and written in standard English?

Reviewer #1: Yes

Reviewer #2: Yes

Reviewer #1: This is an important manuscript describing how one of the world’s most advanced STI surveillance systems has been further enhanced. It is a well-written paper that merits timely publication. I have no major concerns, only a few minor issues and clarifications to suggest.

OVERALL

In the results you highlight: "[In 2023,] transgender women, transgender men, and gender diverse (identifying as non-binary or in any other way) people each made up less than 1% of people accessing SHS in England".

Given the comprehensiveness and reach of the GUMCAD tool, and considering the scarcity of representative estimates for trans and gender-diverse populations, this paper — and the journal's readership — would benefit from more prominent presentation of the following prevalence figures for SHS attendees: transgender women, 0.4%; transgender men, 0.5%; and gender-diverse individuals, 0.5%.

ABSTRACT

- Unless incidence estimates have been published elsewhere, and given the absence of clear evidence for increases in HSV or HPV, I suggest rephrasing to: “...an increasing trend in diagnosed bacterial sexually transmitted infections.”

- Even within the abstract, it would be more accessible to spell out GUMCAD in full upon first mention.

INTRODUCTION

- The opening paragraph comes across as somewhat dramatic. A large proportion of diagnosed bacterial STIs affect the rectum or oropharynx and often resolve spontaneously without leading to the sequelae described. Consider toning this down slightly.

- P3L63: The second part of this sentence — suggesting a greater proportional rise among heterosexuals than MSM — is potentially misleading. For example, a rise from 1 to 2 cases among heterosexuals (100% increase) is proportionally greater than a rise from 5,000 to 6,000 among homosexual men (20%). Unless underpinned with absolute numbers, I suggest deleting the second half of the sentence.

- P4L83: Reference 10 (Stutterheim et al.) reports on HIV prevalence, not "poorer sexual health." Among trans masculine individuals, the increased HIV risk appears primarily linked to behaviour — particularly sex with other men — rather than gender identity per se. I recommend rephrasing:

“Given evidence for higher HIV prevalence among transgender men and particularly transgender women, when compared to the general population (10),...”

METHODS

- P4L90: The sentence is quite long. Consider placing the legal clause (“under Regulation 4 of The Health Service – Control of Patient Information”) in parentheses or splitting it into a new sentence for readability.

- P4L93: The term “Caldicott Advisory Panel” may be unfamiliar to international readers. I suggest adding a brief explanation such as:

“...the UK's Caldicott Advisory Panel (named after Dame Fiona Caldicott).”

- P5L106: The term “lower super output area” misses a word and should be capitalised and spelled out as: “Lower Layer Super Output Area (LSOA)”, which is likely more recognisable internationally.

- P7L167: It would be more accurate to refer to “NHS England’s Data Coordination Board (DCB)” or, if discussing 2023 specifically, “formerly NHS Digital’s DCB”, as NHS Digital was dissolved and merged into NHS England in 2023. Consider clarifying this transition. Also, “NHS England Digital” may not be meaningful to non-UK audiences — best to avoid or explain.

RESULTS

- P8L203 and P10L260: Replace hyphens with en dashes (–) where ranges are indicated (e.g. “57–71%”).

- P10L238: Add “in” before “96%”: “…gender identity was reported in 96% of records...”

- P10L259f: The current wording is ambiguous. It could be read as implying that people with only one recent sexual partner had more STIs than those with multiple partners. If I understand correctly, partner number data were available for only 28% of cases in 2023. Suggested rephrasing: “In absolute numbers, heterosexual or bisexual women accounted for far more screens and attendances than heterosexual men. Where data on recent partner numbers were available, these women and men typically reported only one sexual partner in the past three months. In contrast, among GBMSM — who more frequently reported multiple partners — most screens and attendances involved men with 2–4 or 5+ recent partners.”

- P11L271 and Fig. 1A: The current labelling mixes sexual identity and behaviour. Labelling only MSM by sexual identity while other groups remain behaviourally defined is inconsistent and could be read as stigmatising. A more consistent terminology might be:

-HMSW (heterosexual and other men who have sex exclusively with women)

-HBWSM (heterosexual, bisexual, and other women who have sex with men)

-GBMSM (gay, bisexual, and other men who have sex with men)

-LWSW (lesbian and other women who have sex exclusively with women)

In any case, where “MSW” is defined, clarify that it refers to “men who have sex exclusively with women.”

Since you mention that “women” in the figure include trans women, please also note the implications: for example, trans women with a penis having sex with each other are categorised as WSW. While this classification may be politically correct, it has implications for epidemiological reporting, particularly if the proportion of trans women increases in the future.

- P11L279: Replace the hyphen with an em dash (—).

DISCUSSION

- P12L310: Add “diagnosed” before “bacterial” to reflect the fact that most countries (unlike the UK) lack comprehensive population-level testing data.

- P13L324: See previous point. GBMSM includes gay, bisexual, and other men who have sex with men, many of whom may not identify as gay or bisexual. You could rephrase as:

“MSM were more likely to report multiple sexual partners than MSW, WSM, or WSW,”

or:

“Gay and bisexual men were more likely to report multiple sexual partners than heterosexual men and women or lesbian women.”

- P13L328: Please add “from three anatomical sites” after “...recommended to test for STIs quarterly”, to clarify the testing intensity involved, which impacts diagnosis rates.

- P14L357: It is reassuring to read that the primary aim of using doxyPEP as a public health intervention is targeted at syphilis. Thank you for this.

Reviewer #2: This paper represents important work in enhancing the surveillance system. The aim of the paper is somewhat unclear, however. There is some description of the consultation process to inform the enhancement work, though the detail of this is difficult to follow at time (important detail is left in supporting information). The GUMCAD data presented is limited in terms of responding to the main issues raised in the introduction. Further, the large proportion of missing behavioural data (i.e., gaps in number of recent sex partners) does not seem to support the statements about more robust behavioural data in the discussion.

**Do you want your identity to be public for this peer review?** For information about this choice, including consent withdrawal, please see our Privacy Policy

Reviewer #1: **Yes:** Dr A Jeremias Schmidt

Reviewer #2: No

---

## [Author Response · Author response to Decision Letter 1]

8 Dec 2025

We gratefully receive the Academic Editor and Reviewers' comments and have responded to each of these in the 'Response to Reviewers' document (uploaded separately).

---

## [Editor Report · Decision Letter 1]

4 Jan 2026

Enhancing surveillance of sexually transmitted infections in England with gender identity and behavioural data: the GUMCAD STI Surveillance System

PONE-D-25-23813R1

Dear Dr. Mohammed,

We’re pleased to inform you that your manuscript has been judged scientifically suitable for publication and will be formally accepted for publication once it meets all outstanding technical requirements.

Kind regards,

Sebastian Suarez Fuller, PhD

Academic Editor

PLOS One

---

## [Editor Report · Acceptance letter]

PONE-D-25-23813R1

PLOS One

Dear Dr. Mohammed,

I'm pleased to inform you that your manuscript has been deemed suitable for publication in PLOS One. Congratulations! Your manuscript is now being handed over to our production team.

Kind regards,

on behalf of

Dr. Sebastian Suarez Fuller

Academic Editor

PLOS One